# Shiga Toxin-Producing *Escherichia coli* in Faecal Samples from Wild Ruminants

**DOI:** 10.3390/ani13050901

**Published:** 2023-03-01

**Authors:** Anna Szczerba-Turek, Filomena Chierchia, Piotr Socha, Wojciech Szweda

**Affiliations:** 1Department of Epizootiology, Faculty of Veterinary Medicine, University of Warmia and Mazury in Olsztyn, Oczapowskiego 13, 10-718 Olsztyn, Poland; 2Department of Animal Reproduction with a Clinic, Faculty of Veterinary Medicine, University of Warmia and Mazury in Olsztyn, Oczapowskiego 14, 10-719 Olsztyn, Poland

**Keywords:** wildlife, foodborne disease, STEC, ‘One Health’, infectious disease

## Abstract

**Simple Summary:**

Wildlife is an important source of infectious diseases, including those caused by Shiga toxin-producing Escherichia coli (STEC). STEC are one of the most frequent bacterial agents associated with outbreaks of foodborne disease. This article analyses STEC in faecal samples from red deer and roe deer. The identified O146:H28, O146:HNM, O103:H7, O103:H21, and O45:HNM serotypes and eae/stx2b, stx1a, stx1NS/stx2b, stx2a, stx2b, and stx2g virulence profiles may be potentially pathogenic to humans. The STEC detected in faecal samples from wildlife poses risks to humans, animals, and agricultural production due to the possibility of direct contact with faeces. In conclusion, the pathogenic potential of STEC should be monitored in the context of the ‘One Health’ approach which links human health with animal and environmental health.

**Abstract:**

Wildlife can harbour Shiga toxin-producing *Escherichia coli* (STEC). In the present study, STEC in faecal samples from red deer (*n* = 106) and roe deer (*n* = 95) were characterised. All isolates were non-O157 strains. In red deer, STEC were detected in 17.9% (*n* = 19) of the isolates, and the *eae*/*stx*_2b_ virulence profile was detected in two isolates (10.5%). One STEC strain harboured *stx*_1a_ (5.3%) and eighteen STEC strains harboured *stx_2_* (94.7%). The most prevalent *stx*_2_ subtypes were *stx*_2b_ (*n* = 12; 66.7%), *stx*_2a_ (*n* = 3; 16.7%), and *stx*_2g_ (*n* = 2; 11.1%). One isolate could not be subtyped (NS) with the applied primers (5.6%). The most widely identified serotypes were O146:H28 (*n* = 4; 21%), O146:HNM (*n* = 2; 10.5%), O103:H7 (*n* = 1; 5.3%), O103:H21 (*n* = 1; 5.3%), and O45:HNM (*n* = 1; 5.3%). In roe deer, STEC were detected in 16.8% (*n* = 16) of the isolates, and the *eae*/*stx*_2b_ virulence profile was detected in one isolate (6.3%). Two STEC strains harboured *stx*_1a_ (12.5%), one strain harboured *stx*_1NS_/*stx*_2b_ (6.3%), and thirteen strains harboured *stx_2_* (81.3%). The most common subtypes were *stx*_2b_ (*n* = 8; 61.5%), *stx*_2g_ (*n* = 2; 15.4%), non-typeable subtypes (NS) (*n* = 2; 15.4%), and *stx*_2a_ (*n* = 1; 7.7%). Serotype O146:H28 (*n* = 5; 31.3%) was identified. The study demonstrated that the zoonotic potential of STEC strains isolated from wildlife faeces should be monitored in the context of the ‘One Health’ approach which links human health with animal and environmental health.

## 1. Introduction

Wildlife is an important source of infectious diseases transmitted to humans. It has the potential to harbour foodborne pathogens such as Shiga toxin-producing *Escherichia coli* (STEC), *Campylobacter* spp., *Yersinia enetrocolitica*, or *Salmonella* spp. [1,2,3,4,5,6,7]. Human activities have led to the loss of wildlife habitats and have increased the interactions between wildlife, humans, domestic animals, and livestock. The above has increased the prevalence of zoonotic diseases, and the ‘One Health’ holistic approach was introduced to sustainably regulate and optimise the health of humans, animals, and the ecosystem by meeting the need for clean and nutritious food, water, and air. ‘One Health’ is an interdisciplinary approach involving multiple disciplines, sectors, and communities who work together to develop sustainable natural policies and address the threats to human and ecosystem health. STEC should be addressed under ‘One Health’ research because these pathogens are ubiquitous in various ecosystem niches [8,9,10]. In recent years, the European population of wild ruminants, including red deer *(Cervus elaphus)* and roe deer (*Capreolus capreolus*), has been increasing in density and distribution [2], and information about potential foodborne pathogens carried by these animals is available [1,2,4,11,12,13]. However, little is known about the prevalence of STEC, their pathogenic potential, and the virulence profiles detected in different wildlife species. 

According to the European Food Safety Authority (EFSA), the European Centre for Disease Prevention and Control (ECDC), and the One Health 2020 Zoonoses Report, STEC infections were the fourth most common zoonoses after campylobacteriosis, salmonellosis, and yersiniosis [14,15]. Currently, the categorisation of STEC pathogenicity is based on serotype; STEC belonging to O-group, namely O26, O103, O111, O145, and O157, are termed top-five and used to be the most frequently detected O-group STEC among patients with the haemolytic uraemic syndrome (HUS) [16]. Public health authorities focus mostly on O157 STEC infections due to their pathogenicity, whereas non-O157 STEC serogroups such as O26, O103, O111, O121, and O145 cause twice as many infections in humans [17,18]. According to the EFSA Authority Panel on Biological Hazards (EFSA BIOHAZ), the Food and Agriculture Organization of the United Nations, and the World Health Organization (FAO/WHO), the pathogenic capacity of STEC strains is determined mainly by the number of virulence factors such as *Shiga toxin* 1 (*stx*_1_*), Shiga toxin* 2 (*stx*_2_), *E. coli* attaching and effacing (*eae*), or transcriptional activator of aggregative adherence fimbria I (*aggR*) genes [19,20,21]. STEC strains harbouring the *eae* gene are referred to as attaching–effacing Shiga toxin-producing *E. coli* (AE-STEC). There are at least three variants of the *stx*_1_ gene (*stx*_1a_, *stx*_1c_, and *stx*_1d_) and seven variants of the *stx_2_* gene (*stx*_2a_, *stx*_2b_, *stx*_2c_, *stx*_2d_, *stx*_2e_, *stx*_2f_*,* and *stx*_2g_) [22]; however, new *stx*_2_ variants are being discovered all the time (*stx*_2k_, *stx*_2i_, *stx*_2h_, *stx*_2d1_, and *stx*_2d2_) [23,24]. These variants have been linked to differences in clinical outcomes and toxicity. *stx* variants are also associated with disease severity. Some *stx2* subtypes, such as *stx*_2a_, *stx*_2c_, and *stx*_2d_*,* are frequently linked to a higher risk of HUS, whereas other subtypes, such as *stx*_2e_, *stx*_2b_, *stx*_2f_, and *stx*_2g_, are linked to less severe illnesses [19,21]. STEC strains are transmitted to humans mainly through the consumption of contaminated food products of animal origin [5], but also through vectors (ticks, fleas, mosquitoes, and rodents), scratches or bites, soil and water, direct contact with farmers, abattoir workers, veterinary practitioners, and animal faeces [21]. Wild ruminants are a reservoir of STEC [1,2,5,12,13]. Asymptomatic animals can harbour and shed STEC, including through faeces, thus contaminating the environment, water, and agriculture.

The aim of the present study was to investigate the prevalence, virulence profiles, and pathogenic potential of STEC in faecal samples from red deer and roe deer living in their natural habitats.

## 2. Materials and Methods

### 2.1. Study Area, Sample Collection, and Initial Processing

A total of 201 faecal samples were collected from wild animals including 106 red deer and 95 roe deer between May 2020 and May 2021 in north-eastern Poland (Warmian-Masurian and Podlaskie Voivodeships). Fresh faecal samples were collected in designated feeding locations in the forest. ‘Point 0′ was the observation of faeces upon arrival at the feeding site. From then on, freshly deposited faeces were collected every 3–4 h, always within 24 h of being passed by the target species at each location. Both hunters and foresters have specialised knowledge and are able to correctly identify both faeces and animal species. The samples were kept in a field cooler and stored at 4 °C until processing (within 24 h). In the laboratory, faecal samples of one g each were ground, combined with 10 mL of buffered peptone water (BPW) (BTL, Łódź, Poland) in aseptic conditions, and incubated overnight at 37 °C, with an aeration rate of 180 rpm. The resulting culture was used to extract DNA and isolate STEC. An amount of 500 µL of each enrichment culture was stored at −80 °C in 30% sterile glycerol.

### 2.2. Extraction of DNA and STEC screening by PCR

DNA was extracted from 1 mL of each BPW (BTL, Łódź, Poland) enrichment culture using a Genomic Mini kit (A&A Biotechnology, Gdynia, Poland) according to the manufacturer’s instructions. All samples were tested for *stx*_1_, *stx*_2_, and *eae* using the procedure recommended by the European Union Reference Laboratory for *E. coli* (EU-RL VTEC_Method 01 for *E. coli*) and *aggR* genes described previously [13,25,26,27,28,29,30,31]. An amount of 50 µL of BPW (BTL, Łódź, Poland) enriched cultures of the samples with amplified *stx*_1_, *stx*_2_, *eae* or *aggR* genes were plated on CHROMagar STEC (1381)—a selective medium for the isolation of STEC (GrasoBiotech, Starogard Gdański, Poland), and incubated at 37 °C for approximately 24 h. Mauve colonies were isolated, and the presence of *stx*_1_, *stx*_2_, *eae*, or *aggR* was confirmed by conventional PCR. After confirmation, each STEC isolate was stored at −80 °C in 30% sterile glycerol.

### 2.3. Shiga Toxin Subtyping and Molecular Serotyping

The presence of *stx*_1_ and *stx*_2_ subtypes in the STEC isolates was determined according to the methods described by Scheutz et al. [22] and the EU-RL procedure for *E. coli* (EU-RL VTEC_Method 06) [32]. STEC serogroups associated mainly with human infections (O antigen-encoding genes *wzx*, *wbgN*, *wzy*, and *rfb* specific to thirteen serogroups: O26, O45, O55, O91, O103, O104, O11, O113, O121, O128, O145, O146, and O157) were identified according to the EU-RL procedure for *E. coli* (EU-RL VTEC_Method 03) [33,34]. H antigens encoding *fliC* (specific to flagellar genes) related to H7, H8, H11, H21, and H28 were identified based on the protocols proposed by Durso et al., Gannon et al., and Mora et al. [35,36,37].

### 2.4. Statistical Analysis

The Clopper–Pearson ‘exact’ method based on beta distribution at a significance level of α = 0.05 was used to evaluate the prevalence of STEC strains in red deer and roe deer populations. Statistical comparisons were conducted using EpiTools—free epidemiological calculators [38].

## 3. Results

Fresh faecal samples were collected from 106 red deer. In the red deer population, STEC were detected in 19 isolates (17.92%, 95% CI = 11.15–26.57). All isolates were non-O157 strains, and two isolates belonged to the O103 serogroup (one of the top five serogroups) (10.53%, 95% CI = 1.30–33.14). All isolates were *aggR*-negative. Two isolates were *eae*-positive (10.53%, 95% CI = 1.30–33.14) with the *eae/stx*_2b_ virulence profile, and one of them belonged to the O146 serogroup. The remaining isolates contained the *stx*_1_ gene or the *stx*_2_ gene (89.47%, 95% CI = 66.86–98.70). One STEC harboured *stx*_1a_ (5.26%, 95% CI = 0.13–26.03) and belonged to the O103:H7 serotype. Sixteen isolates harboured only *stx*_2_ (84.21%, 95% CI = 60.42–96.62). *stx*_2b_ was the most prevalent subtype that was detected in 10 isolates (62.50%, 95% CI = 35.43–84.80). Three isolates harboured *stx*_2a_ (18.75%, 95% CI = 4.05–45.65), one isolate harboured *stx*_2g_ (6.25%, 95% CI = 0.16–30.23), and one isolate could not be subtyped with the applied primers (6.25%, 95% CI = 0.16–30.23). O146 was the most prevalent O-group that was detected in six isolates (31.58%, 95% CI = 12.58–56.55); two isolates belonged to the O103 serogroup (10.53%, 95% CI = 1.30–33.14) and one isolate belonged to the O45 serogroup (5.26%, 95% CI = 0.13–26.03). The pathotypes, *stx* subtypes, and serogroups of STEC isolates collected from the red deer population are characterised in Table 1.

Fresh faecal samples were collected from 95 roe deer. In roe deer faeces, STEC were detected in 16 isolates (16.84%, 95% CI = 9.94–25.90). All isolates were non-O157 strains, and none belonged to the O-group (top five serogroups). All isolates were *aggR*-negative. One isolate was *eae*-positive with the *eae/stx*_2b_ virulence profile (6.25%, 95% CI = 0.16–30.23). Two STEC isolates harboured *stx*_1a_ (12.50%; 95% CI = 1.55–38.35), but O-antigen groups could not be identified with the applied primers. One isolate harboured *stx*_1NS_*/stx*_2b_ (6.25%, 95% CI = 00.16–30.23). Twelve isolates harboured only the *stx*_2_ gene (75.00%, 95% CI = 47.62–92.73). *stx*_2b_ was the most prevalent subtype that was detected in seven isolates (58.33%, 95% CI = 27.67–84.83), one isolate harboured *stx*_2a_ (8.33%, 95% CI = 00.21–38.48), two isolates harboured *stx*_2g_ (16.67%, 95% CI = 2.09–48.41), and two isolates could not be subtyped with the applied primers (16.67%, 95% CI = 2.09–48.41). The most prevalent O-group in the roe deer population was O146, which was detected in five isolates (31.25%, 95% CI = 11.02–58.66). The pathotypes, *stx* subtypes, and serogroups of STEC isolates collected from the roe deer population are characterised in Table 2.

## 4. Discussion

In the present study, the prevalence and pathogenic potential of STEC strains isolated from the faeces of 106 red deer and 95 roe deer were examined. Wildlife faeces were analysed in this study because they directly enter the environment, fields, and water. Agricultural products such as vegetables may become contaminated with STEC pathogens through direct or indirect contact with wildlife. STEC strains were isolated from 17.92% of faecal samples collected from red deer and 16.84% of faecal samples collected from roe deer. All STEC strains were serotyped as non-O157 and were *aggR*-negative. Serotypes O146:H28, O146:HNM, O103:H7, O103:H21, and O45:HNM, and the *eae/stx*_2b_, *stx*_1a_, *stx*_1NS_*/stx*_2b_, *stx*_2a_, *stx*_2b_, and *stx*_2g_ virulence profiles were identified. According to the EFSA, ECDC, and the European Union One Health 2020 Zoonoses Report, STEC serogroups (based on the O antigen) O26, O157, O103, O145, O146, O91, O80, and O128 were most commonly identified in humans, whereas *stx*_2a_ and *stx*_2b_ virulence profiles were reported in patients with HUS, bloody diarrhoea, and hospitalised patients. Virulence profile *stx*_1a_ was also identified in patients with bloody diarrhoea and patients requiring hospitalisation [15].

The study demonstrated that red deer and roe deer are potential carriers of non-O-157 STEC isolates that may be pathogenic to humans. This is an important consideration because the red deer population has been increasing steadily in Europe, and wildlife have direct and indirect contact with humans, domestic animals, livestock, water bodies, and the environment [39,40]. The results also confirmed that wildlife, including red deer and roe deer, could play a role as reservoirs and shedders of STEC in the environment [3,40,41,42,43]. The prevalence and pathogenic potential of STEC strains isolated from rectal swabs from red deer and roe deer were examined in our previous study. Four O157:H7 (4.1%) strains were isolated from red deer and one O157:H7 (0.75%) strain was isolated from roe deer. STEC strains were identified in 21.65% and 24.63% of rectal swabs from red deer and roe deer, respectively [13]. The most dangerous human STEC virulence profile, *stx*_2a_ was identified in one STEC strain from roe deer and one strain from red deer [13]. The STEC strains isolated in this study appear to have a lower pathogenic potential than those isolated from rectal swabs in our previous experiment. However, pathogenic potential exists. The environmental spread of STEC should be monitored and controlled, especially since rare *stx* human subtypes, such as *stx*_2g_*,* have been detected in human clinical samples in Denmark and Germany [44,45], and the *stx*_2g_ virulence profile was associated with the outbreak of HUS in France [46].

The results of studies conducted in different European countries indicate that wildlife such as red deer and roe deer could carry mainly STEC belonging to serogroups other than the top five serogroups (O26, O103, O111, O145, and O157). In Italy, the prevalence of STEC in free-ranging red deer was 19.9%, O146:H28 was the most frequently detected serotype (32.3%), and all isolated strains were non-O157:H7 [42]. In Spain, the prevalence of STEC was 24.7% in red deer and 5% in roe deer, O146 was the most frequently detected serogroup, and all strains were non-O157 [1]. However, in another Spanish study, the prevalence of STEC in faecal samples collected from red deer was 35%, four isolates (4.3%) belonged to O157:H7, and the remaining ones were non-O157 (95.7%) [47]. In Portugal, the prevalence of STEC was 9.5% in red deer and 25% in roe deer; all STEC strains were non-O157, and serogroup O146 was the second (after O27) most frequently identified O-group [2]. In Germany, none of the STEC isolated from faecal samples collected from roe deer belonged to the top-five serogroups [48]. In the current study, two strains isolated from faecal samples collected from red deer belonged to group O103 (one of the top five serogroups) with the virulence profiles *stx*_1a_ and *stx*_2g_. These results corroborate the findings of Spanish and Italian authors who found that STEC strains isolated from red deer were mainly *eae*-negative (89.47%), and *stx*_2_ was the most common *stx* type (84.21%) [2,42,47]. Nevertheless, there is a need for more accurate and complete data about the pathogenic potential of STEC isolated from wildlife, in particular red deer and roe deer populations. In this study, STEC strains isolated from roe deer were mainly *eae*-negative (93.75%), and *stx*_2_ was the most common *stx* type (75%). Further research is needed to determine why STEC strains isolated from faecal samples collected from red deer and roe deer seem to have lower pathogenic potential than STEC strains isolated from rectal swabs taken from red deer and roe deer in our previous study [13]. More information is needed to accurately characterise STEC strains present in the environment.

There is no doubt that wildlife can carry STEC that are dangerous to human life and health. Due to the loss of natural wildlife habitats and rapid population growth, wild animals are increasingly observed in fields, near farms, and in other areas that are used by people and other animals. Humans, livestock, and wildlife belong to the same ecosystem, and the health of each should be equally important. Emergent zoonotic bacterial diseases should be identified and controlled under the ‘One Health’ approach, which integrates the efforts of physicians, veterinarians, epidemiologists, microbiologists, public health workers, and hunters. To prevent infections, the food chain should be controlled at all levels, from agricultural production, processing, and food preparation to production facilities and domestic kitchens. Hygiene education is indispensable because environmental strains should be monitored to minimise the risk of infection and outbreaks of foodborne diseases.

## 5. Conclusions

The majority of STEC strains isolated from faecal samples collected from red deer and roe deer did not belong to the top five serogroups. O146 was the most prevalent O-group. All STEC isolates were *aggR*-negative, and most of them were *eae*-negative. *stx*_2_ was the most common *stx* gene type, and *stx*_2b_ was the most common subtype of the *stx*_2_ gene. *stx*_2a_, *stx*_2b_, and *stx*_2g_ virulence profiles were identified, and these profiles have been reported in patients with HUS and bloody diarrhoea. Wild ruminants such as red deer and roe deer are reservoirs of potentially pathogenic STEC. These animals can shed STEC in faeces and contaminate agricultural production systems, water, and the environment. According to the ‘One Health’ concept, naturally occurring STEC strains should be monitored in the environment, agriculture, and water to evaluate the risks to public health. New information about serogroups and virulence profiles is needed to expand our knowledge about the epidemiology and circulation of STEC in the environment. Knowledge and communication are necessary for prevention and control strategies.

## Figures and Tables

**Table 1 animals-13-00901-t001:** Serotype, *stx* subtype, and the number of isolates harbouring virulence genes in STEC strains isolated from red deer (*Cervus elaphus*) faeces.

Pathotypes	Total	*stx* Subtype	Serogroups
STEC *stx*_1_	1	*stx*_1a_ (1)	O103:H7 (1)
STEC *stx*_2_	16	*stx*_2a_ (3) *stx*_2b_ (10) *stx*_2g_ (2) *stx*_NS1_ (1)	ONT^2^:HNM ^3^ (2), O45:HNM ^3^(1) O146:H28 (4), ONT^2^:HNM ^3^ (3), ONT ^2^:H7 (1) ONT ^2^:H21 (1), O146:HNM ^3^ (1), ONT ^2^:H28 (1), O103:H21 (1) ONT ^2^:H28 (1)
AE-STEC *eae/stx*2	2	*stx* _2b_	ONT ^2^:HNM ^3^ (1) O146: HNM ^3^ (1)

^1^ NS, not subtyped with the use of the primers for Shiga-toxin encoding genes (*stx*). ^2^ ONT, O antigen non-typeable. ^3^ HNM, H antigen non-motile.

**Table 2 animals-13-00901-t002:** Serotype, *stx* subtype, and the number of isolates harbouring virulence genes in STEC strains isolated from roe deer (*Capreolus capreolus*).

Pathotypes	Total	*stx* Subtype	Serogroups
STEC *stx*_1_	2	*stx*_1a_ (2)	ONT ^2^:HNM ^3^ (1) ONT ^2^:H21 (1)
STEC *stx*_1_ + *stx*_2_	1	*stx* _1NS*1*_ */stx* _2b_	ONT ^2^:HNM ^3^
STEC *stx*_2_	12	*stx*_2a_ (1) *stx*_2b_ (7) *stx*_2g_ (2) *stx*_NS1_ (2)	ONT ^2^:HNM ^3^ O146:H28 (5), ONT ^2^:H21 (2) ONT ^2^:H28 (1) ONT ^2^:HNM ^3^
AE-STEC *eae/stx*_2_	1	*stx* _2b_	ONT ^2^:HNM ^3^

^1^ NS, not subtyped with the use of the primers for Shiga-toxin encoding genes (*stx*). ^2^ ONT, O antigen non-typeable. ^3^ HNM, H antigen non-motile.

## Data Availability

Not applicable.

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
