# Peer review of "Shiga Toxin-Producing Escherichia coli in Faecal Samples from Wild Ruminants"

_animals, 2023, doi:10.3390/ani13050901_

Round 1
Reviewer 1 Report
The authors of animals-2175418-v1_review have written a paper regarding STEC in faecal samples in two deer species which unfortunately in various parts is far to pretentious and with unclear reasoning.
For example the “Sample Summary” starts a bit strange; If STEC is one of the most frequent bacterial agents associated with foodborne outbreaks why did the authors investigate faecal samples of red and roe deer.
The authors are not up to date with the most recent stx2 variants.
Lines 76-81 are contradicting “The transmission of STEC strains to humans occurs primarily through the consumption of contaminated food products of animal origin” while in the next sentence the authors state “Wild ruminants are the STEC reservoir”. Where I am from, wild ruminants are not common in a humans diet.
Please provide a reference if wild ruminants are considered THE STEC reservoir.
How did the authors check that the faecal samples belonged to the target deer species when accepting 24h old faeces.
It is unclear from the manuscript whether the authors were able to obtain a STEC isolate from all stx PCR positive enrichments.
Please join the results and discussion sections since the discussion (lines 168-187) is more or less a repetition of the results section.
The discussion in lines 199-201 is over de top since the authors consider that with a prevalence of 16.84%-17.92% red and roe deer “could play a significant role as reservoirs and super-200 shedders of STEC in the environment” (lines 200-201). Moreover in their previous study O157:H7 was isolated while in the current manuscript only non-O157 were obtained. This should be discussed in more detail.
Line 212-214 are very hard to follow, since no eae/stx2g positive isolates were recovered from the deer sampled.
The manuscript should be read and corrected by a native English speaking person. It needs rewriting in various parts.
Minor comments
Throughout the manuscript please write the stx variants; stx in italic and the variant in subscript next to it not in italic.
Abstract; please use 1 decimal with the percentages.
Line75; stx2b, stx2f
Line 91; that belonged
Line 105; were plated
Line 150; possible to determine the O-group with the primers used
Line 152; please replace popular with common
Line 154-155; two were not typable with the primers used
Line 166-168; Please rephrase
Author Response
The authors would like to thank the Editor and the Reviewers for a thorough perusal of our manuscript, and for the valuable comments and suggestions that helped us improve the quality of the paper. Below are point-by-point responses to the comments made by Reviewer 1. All content-related changes introduced to the text are highlighted in green, and new text and English corrections are marked in red.
Response to Reviewer 1 Comments
Point 1:
The authors of animals-2175418-v1_review have written a paper regarding STEC in faecal samples in two deer species which unfortunately in various parts is far to pretentious and with unclear reasoning. For example the “Sample Summary” starts a bit strange; If STEC is one of the most frequent bacterial agents associated with foodborne outbreaks why did the authors investigate faecal samples of red and roe deer.
Response 1:
In Europe, wild ruminants are expanding in both range and number, and humans encroach on wild animal habitats, which increases the risk of contact between humans and wild ruminants that may be a source of zoonotic agents such as STEC. It is a well-known fact that ruminants are reservoirs of STEC, but there is little information on STEC prevalence in wild ruminants, especially with regard to the virulence profiles of these strains. The authors studied faecal samples from red deer and roe deer to expand the existing knowledge about naturally occurring STEC strains that may contaminate agricultural products, water and the environment.
Point 2:
The authors are not up to date with the most recent stx2 variants.
Response 2:
Actually, the Authors are up to date with the new subtypes of stx2, stx2k and stx2i, described by Yang et al. (2020) as well as stx2h, stx2d1 and stx2d2, described by Bai et al. (2018), but our methodology is consistent with that proposed by the European Union Reference Laboratory for E. coli. Identification of the subtypes of verocytotoxin encoding genes (vtx) of Escherichia coli by conventional PCR. 2021, Available online. https://www.iss.it/documents/5430402/0/EURL-VTEC_Method_06_Rev+2.pdf/20226358-3792-de68-2f24-1f0249466e0e?t=1619466269431, accessed on 04.05.2022.
If only this methodology is officially modified, we will modify our methodology accordingly.
Point 3:
Lines 76-81 are contradicting “The transmission of STEC strains to humans occurs primarily through the consumption of contaminated food products of animal origin” while in the next sentence the authors state “Wild ruminants are the STEC reservoir”. Where I am from, wild ruminants are not common in a human diet.
Response 3:
Wild ruminants are hunted and consumed in the European Union. A total of 97 wildlife species are hunted in the EU, and 26 bird species and 12 mammalian species are consumed. The most popular food game species are red deer, roe deer, hare (Lepus europaeus), pheasant (Phasianus colchicus) and wild boar (Sus scrofa). Red deer, roe deer and hares are hunted in 17, 16 and 15 EU countries, respectively, whereas pheasants and wild boars are hunted in 14 EU countries. The size and density of wildlife populations are highest in Central Europe, southern Scandinavia and the Baltic countries. On account of its geographical location, Poland is the European leader in terms of forest area which occupies 29.2% of its territory (State Forests official website: http://www.lasy.gov.pl). Poland is one of the largest European producers and exporters of game meat, and annual production is estimated at 12,000 to 14,000 tons. The popularity of game meat, in particular fallow deer and red deer meat, is on the rise among health-conscious consumers due to its high nutritional value, high protein content, low fat content, and unique taste. We addressed the possibility of carcass contamination in our previous article: Szczerba-Turek, A.; Siemionek, J.; Socha, P.; Bancerz-Kisiel, A.; Platt-Samoraj, A.; Lipczynska-Ilczuk, K.; Szweda, W. Shiga toxin-producing Escherichia coli isolates from red deer (Cervus elaphus), roe deer (Capreolus capreolus) and fallow deer (Dama dama) in Poland. Food Microbiol. 2020, 86, 7, doi:10.1016/j.fm.2019.103352.
The present manuscript is more focused on the environment and agriculture.
Point 4:
Please provide a reference if wild ruminants are considered THE STEC reservoir.
Response 4:
The following references were provided: ‘Wild ruminants are the STEC reservoir [1,2,5,12,13].’
- Sanchez, S.; Garcia-Sanchez, A.; Martinez, R.; Blanco, J.; Blanco, J.E.; Blanco, M.; Dahbi, G.; Mora, A.; de Mendoza, J.H.; Alonso, J.M.; et al. Detection and characterisation of Shiga toxin-producing Escherichia coli other than Escherichia coli O157:H7 in wild ruminants. Vet. J. 2009, 180, 384-388, doi:10.1016/j.tvjl.2008.01.011.
- Dias, D.; Caetano, T.; Torres, R.T.; Fonseca, C.; Mendo, S. Shiga toxin-producing Escherichia coli in wild ungulates. Sci. Total Environ. 2019, 651, 203-209, doi:10.1016/j.scitotenv.2018.09.162.
5 Espinosa, L.; Gray, A.; Duffy, G.; Fanning, S.; McMahon, B.J. A scoping review on the prevalence of Shiga-toxigenic Escherichia coli in wild animal species. Zoonoses Public Health 2018, 65, 911-920, doi:10.1111/zph.12508.
- Carrillo-Del Valle, M.D.; De la Garza-Garcia, J.A.; Diaz-Aparicio, E.; Valdivia-Flores, A.G.; Cisneros-Guzman, L.F.; Rosario, C.; Manjarrez-Hernandez, A.H.; Navarro, A.; Xicohtencatl-Cortes, J.; Maravilla, P.; et al. Characterization of Escherichia coli strains from red deer (Cervus elaphus) faeces in a Mexican protected natural area. Eur. J. Wild. Res. 2016, 62, 415-421, doi:10.1007/s10344-016-1015-z.
- Szczerba-Turek, A.; Siemionek, J.; Socha, P.; Bancerz-Kisiel, A.; Platt-Samoraj, A.; Lipczynska-Ilczuk, K.; Szweda, W. Shiga toxin-producing Escherichia coli isolates from red deer (Cervus elaphus), roe deer (Capreolus capreolus) and fallow deer (Dama dama) in Poland. Food Microbiol. 2020, 86, 7, doi:10.1016/j.fm.2019.103352.
Point 5:
How did the authors check that the faecal samples belonged to the target deer species when accepting 24h old faeces.
Response 5:
In Poland, wild animals are fed by foresters and hunters in designated feeding locations in the forest. Animal faeces were collected at such sites. ’Point 0’ was the observation of faeces upon arrival at the feeding site. From then on, freshly deposited faeces were collected every 3-4 h, always within 24 h of being passed by the target species at each location. Both hunters and foresters have specialised knowledge and are able to correctly identify both faeces and animal species.
Point 6:
It is unclear from the manuscript whether the authors were able to obtain a STEC isolate from all stx PCR positive enrichments.
Response 6:
CHROMagar STEC is a selective medium for the isolation of Shiga toxin-producing Escherichia coli (STEC). STEC colonies are very characteristic, and we had no problem with isolation.
Point 7:
Please join the results and discussion sections since the discussion (lines 168-187) is more or less a repetition of the results section.
Response 7:
The fragment in lines 168-187 was removed and replaced with:
Serotypes O146:H28, O146:HNM, O103:H7, O103:H21 and O45:HNM, and eae+/stx2b, eae-/stx1a, eae-/stx1NS/stx2b, eae-/stx2a, eae-/stx2b, eae-/stx2g virulence profiles were identified.
Point 8:
The discussion in lines 199-201 is over de top since the authors consider that with a prevalence of 16.84%-17.92% red and roe deer “could play a significant role as reservoirs and super-200 shedders of STEC in the environment” (lines 200-201). Moreover in their previous study O157:H7 was isolated while in the current manuscript only non-O157 were obtained. This should be discussed in more detail.
Response 8:
The word ‘super’ was removed.
However, STEC strains that are potentially pathogenic to humans were isolated.
Point 9:
Line 212-214 are very hard to follow, since no eae/stx2g positive isolates were recovered from the deer sampled.
Response 9:
We detected two eae-/stx2g positive isolates in red deer and two eae-/stx2g positive isolates in roe deer.
Point 10:
The manuscript should be read and corrected by a native English speaking person. It needs rewriting in various parts.
Response 10:
The entire manuscript was edited to correct grammatical and stylistic errors. Linguistic corrections are marked in red.
Minor comments
Throughout the manuscript please write the stx variants; stx in italic and the variant in subscript next to it not in italic.
The relevant corrections were made throughout the manuscript.
Abstract; please use 1 decimal with the percentages.
The relevant corrections were made throughout the manuscript.
Line75; stx2b, stx2f
The relevant corrections were made throughout the manuscript.
Line 91; that belonged
This sentence was rewritten.
Line 105; were plated
The correction was made.
Line 150; possible to determine the O-group with the primers used
Line 152; please replace popular with common
Line 154-155; two were not typable with the primers used
Line 166-168; Please rephrase
All suggested corrections were made.
Once again, we would like to thank the Reviewer for all valuable comments.

Reviewer 2 Report
Regarding to the current form of the manuscript, I have the following questions & comments:
1. Please correct & improve English language, found numerous spelling errors and bit awkward sentences.
2. What is the distinct novelty of this work? It is not very clearly indicated.
3. In discussion, the first paragraph is describing the results, should move & merge them into result section, or make it simpler and more conclusive.
4. Regarding the different findings (non-O157 vs. O157), between fresh fecal samples (in current study) vs. previous study using rectal swabs, is there any possible reasons can be argued? F.exp. from different sampling areas/zones, seasons, health conditions, etc.? Better elaboration is needed.
5. Conclusion needs be better formulated & strenghtened, make it more concise and neat.
Author Response
The authors would like to thank the Editor and the Reviewers for a thorough perusal of our manuscript, and for the valuable comments and suggestions that helped us improve the quality of the paper. Below are point-by-point responses to the comments made by Reviewer 2. All content-related changes introduced to the text are highlighted in yellow, and new text and English corrections are marked in red.
Response to Reviewer 2 Comments
Point 1:
Please correct & improve English language, found numerous spelling errors and bit awkward sentences.
Response 1:
The entire manuscript was edited to correct grammatical and stylistic errors. Linguistic corrections are marked in red.
Point 2:
What is the distinct novelty of this work? It is not very clearly indicated.
Response 2:
The novelty of this work lies in the characterisation of STEC strains, especially the virulence profiles of STEC isolated from the faeces of red deer and roe deer.
Point 3:
In discussion, the first paragraph is describing the results, should move & merge them into result section, or make it simpler and more conclusive.
Response 3:
This fragment was removed.
Point 4:
Regarding the different findings (non-O157 vs. O157), between fresh fecal samples (in current study) vs. previous study using rectal swabs, is there any possible reasons can be argued? F.exp. from different sampling areas/zones, seasons, health conditions, etc.? Better elaboration is needed.
Response 4:
The fact that O157 was detected in rectal swabs but not in faecal samples is difficult to explain and interpret. It would be interesting to analyse the carcasses and meat of the examined animals. The sampling area was similar but not the same, rectal swabs were taken from hunter-harvested animals, faecal samples were collected in the feeding site, and seasons were similar. We will address this issue in our future study.
Point 5: Conclusion needs be better formulated & strenghtened, make it more concise and neat.
Response 5:
The Conclusion section was modified.

Round 2
Reviewer 1 Report
Reviewers response to Response 1;
Acceptable
Reviewers response to Response 2;
The newest stx2 subtypes should be included in the introduction in some way, because it now reads as if there are only seven variants and that is not true.
My issues are not with the PCR protocols you have used, they are still okay because they are the official ones at the moment.
Reviewers response to Response 3;
Acceptable
Reviewers response to Response 4;
Acceptable
Reviewers response to Response 5;
Acceptable, thank you for explaining the procedure in more detail
Reviewers response to Response 6;
Based on the response I now understand that the authors were successful in isolating a STEC isolate from every stx PCR positive enrichments. So a 100% isolation success rate, please address this in the manuscript, because this is valuable information.
A positive stx PCR tests does not automatically mean the presence of STEC, because a Shiga toxin gene can also occur on a free-living bacteriophage.
Reviewers response to Response 7;
Acceptable
Reviewers response to Response 8;
I still do not agree with the sentence that “The results also confirmed that wildlife, including red deer and roe deer, could play a significant role as reservoirs and shedders of STEC in the environment”, when the prevalence is smaller compared to their previous study.
Reviewers response to Response 9;
Indeed I was incorrect with my comment that the authors did not isolate eae/stx2g positives. But I now realize why I thought that. I continuously overlook the minus after eae. So for me the use of eae- is rather confusing, especially when it is not also included in the Pathotypes columns in Table 1 and Table 2 and with such a low prevalence of eae found...
I would suggest to remove the eae- throughout the manuscript and only use eae in the STEC isolates positive for this gene. The authors also do not add a + after each stx gene/variant.
Reviewers response to Response 10;
Acceptable
Nearly all my minor comments were all addressed except for the correct way in writing the stx variants. To my opinion it should be stx1a, stx1c etc, and stx2a, stx2b etc. So stx in italic, and 1, 1a, 1c etc, 2, 2a, 2b etc in subscript.
Author Response
The authors would like to thank the Editor and Reviewers for a thorough perusal of our manuscript, and for the valuable comments and suggestions that helped us improve the quality of the paper. Below are point-by-point responses to the comments made by Reviewer 1. All content-related changes introduced to the text are highlighted in purple
Reviewers response to Response 1;
Acceptable
Reviewers response to Response 2;
The newest stx2 subtypes should be included in the introduction in some way, because it now reads as if there are only seven variants and that is not true.
My issues are not with the PCR protocols you have used, they are still okay because they are the official ones at the moment.
Response:
We added sentence and refernces“], however, the new stx2 variants have been discovered all the time (stx2k, stx2i, stx2h,stx2d1,stx2d2) [23,24].
- Yang, X.; Bai, X.; Zhang, J.; Sun, H.; Fu, S.; Fan, R.; He, X.; Scheutz, F.; Matussek, A.; Xiong, Y. Escherichia coli strains producing a novel Shiga toxin 2 subtype circulate in China. Int. J. Med. Microbiol. 2020, 310, 151377. doi: 10.1016/j.ijmm.2019.151377.
- Bai, X.; Fu, S.; Zhang, J.; Fan, R.; Xu, Y.; Sun, H.; He, X.; Xu, J.; Xiong, Y. Identification and pathogenomic analysis of an Escherichia coli strain producing a novel Shiga toxin 2 subtype. Sci. Rep. 2018, 8, 6756. doi: 10.1038/s41598-018-25233-x.
Reviewers response to Response 3;
Acceptable
Reviewers response to Response 4;
Acceptable
Reviewers response to Response 5;
Acceptable, thank you for explaining the procedure in more detail
Reviewers response to Response 6;
Based on the response I now understand that the authors were successful in isolating a STEC isolate from every stx PCR positive enrichments. So a 100% isolation success rate, please address this in the manuscript, because this is valuable information.
A positive stx PCR tests does not automatically mean the presence of STEC, because a Shiga toxin gene can also occur on a free-living bacteriophage.
Response:
Thank You for valuable comments
Reviewers response to Response 7;
Acceptable
Reviewers response to Response 8;
I still do not agree with the sentence that “The results also confirmed that wildlife, including red deer and roe deer, could play a significant role as reservoirs and shedders of STEC in the environment”, when the prevalence is smaller compared to their previous study.
Response:
The Authors changed the sentence “The results also confirmed that wildlife, including red deer and roe deer, could play a role as reservoirs and shedders of STEC in the environment”
Reviewers response to Response 9;
Indeed I was incorrect with my comment that the authors did not isolate eae/stx2g positives. But I now realize why I thought that. I continuously overlook the minus after eae. So for me the use of eae- is rather confusing, especially when it is not also included in the Pathotypes columns in Table 1 and Table 2 and with such a low prevalence of eae found...
I would suggest to remove the eae- throughout the manuscript and only use eae in the STEC isolates positive for this gene. The authors also do not add a + after each stx gene/variant.
Response:
All suggested corrections were done
Reviewers response to Response 10;
Acceptable
Nearly all my minor comments were all addressed except for the correct way in writing the stx variants. To my opinion it should be stx1a, stx1c etc, and stx2a, stx2b etc. So stx in italic, and 1, 1a, 1c etc, 2, 2a, 2b etc in subscript.
Response:
All suggested corrections were done
